# Simulation and Optimization of Hemispherical Resonator’s Equivalent Bottom Angle for Frequency-Splitting Suppression

**DOI:** 10.3390/mi14091686

**Published:** 2023-08-29

**Authors:** Zhiyong Gao, Shang Wang, Zhi Wang, Xukai Ding

**Affiliations:** 1School of Fundamental Physics and Mathematical Sciences, Hangzhou Institute for Advanced Study, University of Chinese Academy of Sciences (UCAS), Hangzhou 310012, China; gaozhiyong21@mails.ucas.ac.cn; 2University of Chinese Academy of Sciences, Beijing 101408, China; 3Changchun Institute of Optics, Fine Mechanics and Physics, Chinese Academy of Sciences, Changchun 130033, China; ws790402497@163.com; 4School of Instrument Science and Engineering, Southeast University, Nanjiing 210096, China; ding.xk@seu.edu.cn

**Keywords:** hemispherical resonator, frequency splitting, structural optimization, 4-antinodes vibration mode, mass sensitivity factor

## Abstract

As an inertial sensor with excellent performance, the hemispherical resonator gyro is widely used in aerospace, weapon navigation and other fields due to its advantages of high precision, high reliability, and long life. Due to the uneven distributions of material properties and mass of the resonator in the circumferential direction, the frequencies of the two 4-antinodes vibration modes (operational mode) of resonator in different directions are different, which is called frequency splitting. Frequency splitting is the main error source affecting the accuracy of the hemispherical resonator gyro and must be suppressed. The frequency splitting is related to the structure of the resonator. For the planar-electrode-type hemispherical resonator gyro, in order to suppress the frequency splitting from the structure, improve the accuracy of the hemispherical resonator gyro, and determine and optimize the equivalent bottom angle parameters of the hemispherical resonator, this paper starts from the thin shell theory, and the 4-antinodes vibration mode and waveform precession model of the hemispherical resonator are researched. The effect of the equivalent bottom angle on the 4-antinodes vibration mode frequency value under different boundary conditions is theoretically analyzed and simulated. The simulation results show that the equivalent bottom angle affects the 4-antinodes vibration mode of the hemispherical resonator through radial constraints. The hemispherical resonator with mid-surface radius R=15 mm and shell thickness h=1 mm is the optimization object, and the stem diameter *D* and fillet radius $R_{1}$ are experimental factors, with the 4-antinodes vibration mode frequency value and mass sensitivity factor as the response indexes. The central composite design is carried out to optimize the equivalent bottom angle parameters. The optimized structural parameters are: stem diameter D=7 mm, fillet radii $R_{1}=1\;\mathrm{mm}$, $R_{2}=0.8\;\mathrm{mm}$. The simulation results show that the 4-antinodes vibration mode frequency value is 5441.761 Hz, and the mass sensitivity factor is 3.91 Hz/mg, which meets the working and excitation requirements wonderfully. This research will provide guidance and reference for improving the accuracy of the hemispherical resonator gyro.

## 1. Introduction

The hemispherical resonator gyro (HRG) is a type of Coriolis vibrating gyro that utilizes the standing wave precession effect of vibration of the quartz hemispherical shell to detect rotation without the need for high-speed rotors or movable supports [1]. Its exceptional advantages make it a valuable sensor for a wide range of applications, including the stable control of spacecraft and satellites, precise pointing, spacecraft navigation, oil drilling exploration, weapons, aviation, and navigation [2]. Compared to other forms of gyros such as laser gyro and fiber optic gyro, HRG offers several benefits, including high precision, high reliability, small size, low power consumption, low noise, high stability, long life, and high resistance to radiation. It is considered a preferred “high-value sensor” due to these above significant advantages [3].

The United States is the first country to study HRG. The force-balanced hemispherical resonant gyro developed by Northrop Grumman for the precise pointing of the Hubble Telescope has a remarkable bias stability of 0.00008°/h and an angle random walk of 0.00001°/ $\sqrt{\mathrm{Hz}}$  during the test phase, which are currently the highest reported performance indexes for HRG worldwide [4]. There are two working modes of HRG: the force balance mode and whole-angle mode. The force-balanced HRG has high precision, while its measurement range is small. The whole-angle mode, however, overcomes this drawback. SAFRAN in French has developed a planar-electrode-type HRG and inertial navigation unit, SkyNaute, with a simple structure and assembly for the whole-angle mode. Currently, they have the mass production capacity of HRG with an accuracy range of 0.0005 to 0.005°/h [5].

The quartz hemispherical resonator is the primary component of HRG and is a complex and delicate three-dimensional structure consisting of a hard, brittle, and thin spherical shell. Comprising an inner and outer spherical surface and a stem, the structural parameters of the hemispherical resonator, particularly regarding the stem and the transition fillet, are essential in determining the equivalent bottom angle. However, due to the confidentiality of the technology, there are limited detailed introductions to the structural parameters of the hemispherical resonator, making it challenging to develop an optimization model for simulation and structural optimization.

For example, Xu et al. [6] established an optimization model to evaluate the impact of structural parameters on the vibration characteristics of the hemispherical resonator, and an ideal structural parameter was identified with a shell thickness of 0.9667 mm and a stem diameter of 5.03 mm. However, this model did not consider the fillet parameter, and its response index was single, rendering it insufficient for accurately assessing optimization effects. Huang et al. [7] improved the artificial bee colony algorithm by taking the mass of the hemispherical resonator as a response index, and the hemispherical resonator adapted to the spherical electrodes as the optimization object. However, their optimization model failed to consider the frequency difference between the 4-antinodes vibration mode and the later mode. Finally, Hu et al. [8] attempted to address the optimization of the hemispherical resonator by designing a variable shell thickness hemispherical resonator for the planar-electrode-type HRG and analyzing the influence of its structural parameters on the 4-antinodes vibration mode frequency. However, their research did not identify optimal structural parameters, and the variable shell thickness hemispherical resonator’s manufacturing difficulties limit practical applications.

In summary, when aiming to produce a hemispherical resonator, precise parameters for designing the resonator prove elusive, save for the constraints imposed by the manufacturing process. Consequently, in advancing the development of the hemispherical resonator gyro, the identification of suitable resonator parameters within specific limitations becomes pivotal.

Frequency splitting is the main error affecting the accuracy of HRG [9,10], and the structural parameters of the hemispherical resonator play a significant role in it. To improve the accuracy of the HRG, suppress frequency splitting, and optimize the structural parameters of the hemispherical resonator’s equivalent bottom angle, this paper analyzes the influence of structural parameters of the hemispherical resonator on the 4-antinodes vibration mode frequency value and mass sensitivity factor, theoretically. The study focuses on a classic hemispherical resonator with a radius of R=15 mm and a shell thickness of h=1 mm, centering discussions on the equivalent bottom angle and analyzing the influence of the angle on the 4-antinodes vibration mode frequency value and mass sensitivity factor. The study selects the stem diameter and two fillet radii of the resonator for single-factor analysis, and it takes the stem diameter *D* and fillet $R_{1}$ as the experimental factors and the 4-antinodes vibration mode frequency value and mass sensitivity factor as response indexes. Finally, the central composite design is conducted, parameters of the equivalent bottom angle are determined and optimized, and simulations are carried out, aiming to provide guidance and reference for suppressing the frequency splitting and improving the accuracy of the HRG.

## 2. Dynamic Model of Hemispherical Resonator

### 2.1. Thin Shell Theory

The HRG operates on the 4-antinodes standing wave precession characteristic of the hemispherical resonator. The hemispherical resonator is a unique structure of the hemispherical shell, with a shell thickness much smaller than the radius of the mid-surface. Hence, the dynamic model of the hemispherical resonator is established based on thin shell theory. Assuming the uniformity, continuity, and isotropy of material and that the displacement is much smaller than the shell thickness, the stress and strain conform to Hooke’s law. The thin shell theory relies on two basic Kirchhoff–Love assumptions [11]. First, after deformation, every point remains on the same normal of the deformed mid-surface. Second, the distance between every point remains unchanged, and the normal stress on the surface parallel to the mid-surface can be omitted.

As shown in Figure 1a, let us consider a point *P* on the mid-surface of the shell. $e_{1}$, $e_{2}$, $e_{3}$ are the unit tangent vectors and normal vector along the pairwise orthogonal surface coordinate system α, β, γ. $P^{\prime }$ is the point after the deformation of *P*. The deformation at any point on a thin shell, considering the bending deformation, can be calculated using equations from Kirchhoff–Love theory [12]:(1)$$\begin{array}{cc} \; & \varepsilon_{\alpha }^{z}=\varepsilon_{\alpha }+\kappa_{1}z\\ \; & \varepsilon_{\beta }^{z}=\varepsilon_{\beta }+\kappa_{2}z\\ \; & \gamma_{\alpha \beta }^{z}=\gamma_{\alpha \beta }+2\chi z \end{array}$$
where
(2)$$\begin{array}{cc} \; & \varepsilon_{\alpha }=\frac{1}{A}\frac{\partial u}{\partial \alpha }+\frac{v}{AB}\frac{\partial A}{\partial \beta }+\frac{w}{R_{1}}\;\\ \; & \varepsilon_{\beta }=\frac{1}{B}\frac{\partial v}{\partial \beta }+\frac{u}{AB}\frac{\partial B}{\partial \alpha }+\frac{w}{R_{2}}\;\\ \; & \gamma_{\alpha \beta }=\frac{B}{A}\frac{\partial }{\partial \alpha }\left(\frac{v}{B}\right)+\frac{A}{B}\frac{\partial }{\partial \beta }\left(\frac{u}{A}\right)\;\\ \; & \kappa_{1}=\frac{1}{A}\frac{\partial }{\partial \alpha }\left(\frac{u}{R_{1}}\right)+\frac{1}{AB}\frac{\partial A}{\partial \beta }\frac{v}{R_{2}}-\frac{1}{A}\frac{\partial }{\partial \alpha }\left(\frac{1}{A}\frac{\partial w}{\partial \alpha }\right)-\frac{1}{AB^{2}}\frac{\partial A}{\partial \beta }\frac{\partial w}{\partial \beta }\;\\ \; & \kappa_{2}=\frac{1}{B}\frac{\partial }{\partial \beta }\left(\frac{v}{R_{2}}\right)+\frac{1}{AB}\frac{\partial B}{\partial \alpha }\frac{u}{R_{1}}-\frac{1}{B}\frac{\partial }{\partial \beta }\left(\frac{1}{B}\frac{\partial w}{\partial \beta }\right)-\frac{1}{A^{2}B}\frac{\partial B}{\partial a}\frac{\partial w}{\partial \alpha }\;\\ \; & \chi =\frac{1}{R_{1}}\frac{A}{B}\frac{\partial }{\partial \beta }\left(\frac{u}{A}\right)+\frac{1}{R_{2}}\frac{B}{A}\frac{\partial }{\partial \alpha }\left(\frac{v}{B}\right)-\frac{1}{AB}\left(\frac{\partial^{2}w}{\partial \alpha \partial \beta }-\frac{1}{A}\frac{\partial A}{\partial \beta }\frac{\partial w}{\partial \alpha }-\frac{1}{B}\frac{\partial B}{\partial \alpha }\frac{\partial w}{\partial \beta }\right) \end{array}$$

$\varepsilon_{\alpha }$, $\varepsilon_{\beta }$, and $\gamma_{\alpha \beta }$ are strains along the α, β, and γ direction at point *P*, respectively. κ is the change of curvature of the mid-surface. χ is the torsional deformation of the mid-surface. *u*, *u*, *w* are the projection distance along $e_{1}$, $e_{2}$, and $e_{3}$ respectively. *A* and *B* are Lame coefficients. $R_{1}$ and $R_{2}$ are the principal radii of the curvature at point *P*. *z* is the distance between any point on a thin shell and the min-surface along the direction of axis γ.

For a hemispherical shell, if α and β are spherical coordinates, and we replace α and β with θ and φ, the mid-surface radius is denoted by $R_{0}$. The Lame coefficients of the spherical shell in the spherical coordinates are denoted by $A=B=R_{0}$. Substituting them into Equation (Equation 2), we obtain:(3)$$\begin{array}{cc} \; & \varepsilon_{\theta }=\frac{1}{R_{0}}\left(w+\frac{\partial u}{\partial \theta }\right)\;\\ \; & \varepsilon_{\varphi }=\frac{1}{R_{0}\sin\;\theta }\left(\frac{\partial v}{\partial \varphi }+u\cos\theta +w\sin\theta \right)\;\\ \; & \gamma_{\theta \varphi }=\frac{1}{R_{0}\sin\;\theta }\left(\frac{\partial v}{\partial \theta }\sin\theta -v\cos\theta +\frac{\partial u}{\partial \varphi }\right)\;\\ \; & \kappa_{1}=\frac{1}{R_{0}^{2}}\left(\frac{\partial u}{\partial \theta }-\frac{\partial^{2}w}{\partial \theta^{2}}\right)\;\\ \; & \kappa_{2}=\frac{1}{R_{0}^{2}\sin\;\theta }\left(\frac{\partial v}{\partial \varphi }-\frac{1}{\sin\theta }\frac{\partial^{2}w}{\partial \varphi^{2}}+u\cos\theta -\frac{\partial w}{\partial \theta }\cos\theta \right)\;\\ \; & \chi =\frac{1}{R_{0}^{2}\sin\;\theta }\left(-\frac{\partial^{2}w}{\partial \theta \partial \varphi }+\frac{\partial w}{\partial \varphi }\cot\theta +\frac{\partial u}{\partial \varphi }+\frac{\partial v}{\partial \theta }\sin\theta -v\cos\theta \right) \end{array}$$

### 2.2. Kinetic Equations Based on Lagrangian Method

When the hemispherical resonator undergoes free vibration, assuming that the shell’s mid-surface is incompressible, the three tangential strain components are equal to zero, that is, $\varepsilon_{\theta }=\varepsilon_{\varphi }=0$, $\gamma_{\theta \varphi }=0$. The displacement of each point on the hemispherical resonator is expanded according to the second-order natural vibration mode of an incompressible thin shell [13]:(4)$$\begin{array}{c} u(\theta ,\varphi ,t)=U(\theta )[p(t)\cos(2\varphi )+q(t)\sin(2\varphi )]\\ v(\theta ,\varphi ,t)=V(\theta )[p(t)\sin(2\varphi )-q(t)\cos(2\varphi )]\\ w(\theta ,\varphi ,t)=W(\theta )[p(t)\cos(2\varphi )+q(t)\sin(2\varphi )] \end{array}$$
where U(θ), V(θ), and W(θ) are gain functions of the vibration amplitude along the θ, φ, and *R* directions, respectively. p(t) and q(t) are undetermined vibration functions, including the frequency and phase information of the vibration of the hemispherical resonator, which is hereafter abbreviated as *p* and *q*. To account for the equivalent bottom angle brought about by the stem and fillets, we can substitute Equation (Equation 4) into Equation (Equation 3), which results in [14]:(5)$$\begin{array}{c} U\left(\theta \right)=-\sin\theta \left(\tan^{2}\left(\frac{\theta }{2}\right)+\delta_{0}\cot^{2}\left(\frac{\theta }{2}\right)\right)\\ V\left(\theta \right)=-\sin\theta \left(\tan^{2}\left(\frac{\theta }{2}\right)-\delta_{0}\cot^{2}\left(\frac{\theta }{2}\right)\right)\\ W\left(\theta \right)=\left(2+\cos\theta \right)\tan^{2}\left(\frac{\theta }{2}\right)-\delta_{0}\left(2-\cos\theta \right)\cot^{2}\left(\frac{\theta }{2}\right) \end{array}$$
where $\delta_{0}$ is the ratio between constants of integration and can be obtained from the three different boundary conditions: u=0, or v=0, or w=0 at $\theta =\theta_{0}$, respectively. For each boundary condition, $\delta_{0}$ is determined as follows:(6)$$\begin{array}{cc} \; & \delta_{0}=\pm \tan^{4}\left(\frac{\theta_{0}}{2}\right),\left\{\begin{array}{c} -,u=0\;\mathrm{at}\;\theta =\theta_{0}\\ +,v=0\;\mathrm{at}\;\theta =\theta_{0} \end{array}\right.\;\\ \; & \delta_{0}=\frac{2\;+\;\cos\theta_{0}}{2\;-\;\cos\theta_{0}}\tan^{4}\left(\frac{\theta_{0}}{2}\right),w=0\;\mathrm{at}\;\theta =\theta_{0} \end{array}$$

For the discussion of $\delta_{0}$, detailed information has already been presented in reference [14], so we will refrain from providing further elaboration here.

As shown in Figure 1b, we assume that the radius of the mid-surface of the hemispherical resonator is *R*, the density is ρ, and Poisson’s ratio is μ. θ and φ are the latitude angle and longitude angle, respectively; the origin of the Cartesian coordinate system coincides with the center of the hemispherical resonator, and the coordinates of a point deformed on the mid-surface can be expressed as:(7)$$r^{\prime }=r+\Delta =u\hat{\theta }+v\hat{\varphi }+\left(w+R\right)\hat{r}$$

In the 4-antinodes vibration mode, the absolute acceleration of point *P* is [15]:(8)$$V_{p}=\frac{dr^{\prime }}{dt}+\Omega \times r^{\prime }=\left({\dot{u}}_{t}-\Omega v\cos\theta \right)\hat{\theta }+\left[{\dot{v}}_{t}+\Omega \left(u\cos\theta +\left(R+w\right)\sin\theta \right)\right]\hat{\varphi }+\left({\dot{w}}_{t}-\Omega v\sin\theta \right)\hat{r}$$
where the dot · and subscript *t* mean the first derivative with respect to time *t*. Ω is the angular rate input value. The kinetic energy of the hemispherical shell can be expressed as:(9)$$E_{k}=\frac{1}{2}\rho \int_{V_{h}}dV_{h}V_{p}{}^{2}=\frac{1}{2}hR^{2}\rho \int_{0}^{2\pi }\int_{\theta_{0}}^{\frac{\pi }{2}}V_{p}^{2}\sin\theta d\theta d\varphi$$

Therefore, the kinetic energy can be expressed as:(10)$$\begin{array}{c} E_{k}=\frac{1}{2}\rho hR^{2}\int_{0}^{2\pi }\int_{\theta_{0}}^{\frac{\pi }{2}}\left\{\begin{array}{c} \left({\dot{u}}_{t}^{2}+{\dot{v}}_{t}^{2}+{\dot{w}}_{t}^{2}\right)+2\Omega [\left(u{\dot{v}}_{t}-{\dot{u}}_{t}v\right)\cos\theta \\ +\left({\dot{v}}_{t}\left(R+w\right)-v{\dot{w}}_{t}\right)\sin\theta ]\\ +\Omega^{2}[v^{2}+u^{2}\cos^{2}\theta +w^{2}\sin^{2}\theta \\ +2Rw\sin^{2}\theta +2u\left(R+w\right)\cos\theta \sin\theta ]\\ +\Omega^{2}R^{2}\sin^{2}\theta \end{array}\right\}\sin\theta d\theta d\varphi \end{array}$$

Substituting Equation (Equation 4) into Equation (Equation 10), the kinetic energy of the hemispherical resonator is obtained:(11)$$E_{k}=\frac{1}{2}m_{0}\left({\dot{p}}^{2}+{\dot{q}}^{2}\right)+\frac{1}{2}m_{1}\left(\dot{p}q-p\dot{q}\right)\Omega +\frac{1}{2}m_{2}R^{2}\Omega^{2}+\frac{1}{2}m_{3}\left(p^{2}+q^{2}\right)\Omega^{2}$$
where
(12)$$\begin{array}{cc} \; & m_{0}=\pi \rho hR^{2}\int_{\theta_{0}}^{\frac{\pi }{2}}\left(U^{2}+V^{2}+W^{2}\right)\sin\theta d\theta \;\\ \; & m_{1}=4\pi \rho hR^{2}\int_{\theta_{0}}^{\frac{\pi }{2}}V\left(U\cos\theta +W\sin\theta \right)\sin\theta d\theta \;\\ \; & m_{2}=2\pi \rho hR^{2}\int_{\theta_{0}}^{\frac{\pi }{2}}\sin^{3}\theta d\theta \;\\ \; & m_{3}=\pi \rho hR^{2}\int_{\theta_{0}}^{\frac{\pi }{2}}\left[V^{2}+{\left(U\cos\theta +W\sin\theta \right)}^{2}\right]\sin\theta d\theta \end{array}$$

The potential energy of the hemispherical shell can be expressed as:(13)$$E_{p}=\frac{Eh^{3}}{24\left(1-\mu^{2}\right)}\int_{0}^{2\pi }\int_{\theta_{0}}^{\frac{\pi }{2}}\left[k_{1}^{2}+k_{2}^{2}+2\mu k_{1}k_{2}+2\left(1-\mu \right)\chi^{2}\right]R^{2}\sin\theta d\theta d\varphi$$

Substituting Equation (Equation 3) into Equation (Equation 13), the potential energy of the hemispherical resonator is:(14)$$E_{p}=\frac{1}{2}k_{0}\left(p^{2}+q^{2}\right)$$
where
(15)$$\begin{array}{c} k_{0}=\frac{Eh^{3}\pi }{12R^{2}\left(1-\mu^{2}\right)}\int_{\theta_{0}}^{\frac{\pi }{2}}\left\{\begin{array}{c} {\left({\dot{U}}_{\theta }-{\ddot{W}}_{\theta \theta }\right)}^{2}+\frac{1}{\sin^{2}\theta }{\left(2V+\frac{1}{\sin\theta }4W+U\cos\theta -{\dot{W}}_{\theta }\cos\theta \right)}^{2}\\ +2\mu \frac{1}{\sin\theta }\left({\dot{U}}_{\theta }-{\ddot{W}}_{\theta \theta }\right)\left(2V+\frac{1}{\sin\theta }4W+U\cos\theta -{\dot{W}}_{\theta }\cos\theta \right)\\ +2\left(1-\mu \right)\frac{1}{\sin^{2}\theta }{\left(2{\dot{W}}_{\theta }-2W\cot\theta -2U+{\dot{V}}_{\theta }\sin\theta -V\cos\theta \right)}^{2} \end{array}\right\}\sin\theta d\theta \end{array}$$

The Lagrangian equations for the hemispherical resonator can be constructed in case of undamped free vibration:(16)$$\begin{array}{c} L=E_{k}-E_{p}\\ \frac{d}{dt}\left(\frac{\partial L}{\partial \dot{p}}\right)-\frac{\partial L}{\partial p}=0\\ \frac{d}{dt}\left(\frac{\partial L}{\partial \dot{q}}\right)-\frac{\partial L}{\partial q}=0 \end{array}$$

Therefore, the kinetic equations of the hemispherical resonator are obtained as follows:(17)$$\begin{array}{cc} \; & \ddot{p}+2k_{1}\Omega \dot{q}+\left(\omega_{0}^{2}-2k_{2}\Omega^{2}\right)p+k_{1}\dot{\Omega }q=0\;\\ \; & \ddot{q}-2k_{1}\Omega \dot{p}+\left(\omega_{0}^{2}-2k_{2}\Omega^{2}\right)q-k_{1}\dot{\Omega }p=0 \end{array}$$
where $2k_{1}=\frac{m_{1}}{m_{0}}$, $2k_{2}=\frac{m_{3}}{m_{0}}$, $\omega^{2}=\frac{k_{0}}{m_{3}}$.

### 2.3. Dynamics Simulation

z=p+iq is introduced when considering $\dot{\Omega }=0$; then, Equation (Equation 17) can be expressed as
(18)$$\ddot{z}-2k_{1}\Omega i\dot{z}+\left(\omega_{0}^{2}-2k_{2}\Omega^{2}\right)z=0$$

The solution of Equation (Equation 18) is defined as
(19)$$z\left(t\right)=e^{ik_{1}\Omega t}\left(C_{1}e^{-i\omega_{n}t}+C_{2}e^{i\omega_{n}t}\right)$$
where $C_{1}$ and $C_{2}$ are undetermined coefficients, $\omega_{n}=\sqrt{\omega_{0}^{2}+\left(k_{1}^{2}-2k_{2}^{2}\right)\Omega^{2}}$. In the operation of a planar-electrode-type HRG, the sensing mechanism relies on the change in capacitance formed by the end face of the hemispherical resonator’s lip and planar electrodes. The vibration in the busbar direction at the end face of the resonator’s lip is expressed as:(20)$$\begin{array}{cc} u(\varphi ,t) & =U(\pi /2)[p(t)\cos(2\varphi )+q(t)\sin(2\varphi )]\\ \; & =2U\left(\pi /2\right)\left[b\cdot \cos\left(2\varphi -k_{1}\Omega t\right)-a\cdot \sin\left(2\varphi -k_{1}\Omega t\right)\right]\cdot \sin\left(\omega_{n}t\right)\\ \; & =2U\left(\pi /2\right)\sqrt{a^{2}+b^{2}}\cos2\left(\varphi -K\Omega t+\varphi_{0}\right)\sin\left(\omega_{n}t\right) \end{array}$$
where ${z\left(t\right)|}_{t=0}=0$, $C_{1}=-C_{2}=a+ib$, $\varphi_{0}=\frac{1}{2}\arctan\frac{a}{b}$.

Bryan’s factor is
(21)$$K=\frac{k_{1}}{2}=\frac{m_{1}}{4m_{0}}=\frac{\int_{\theta_{0}}^{\frac{\pi }{2}}V\left(U\cos\theta +W\sin\theta \right)\sin\theta d\theta }{\int_{\theta_{0}}^{\frac{\pi }{2}}\left(U^{2}+V^{2}+W^{2}\right)\sin\theta d\theta }$$

Assuming that the effect of the equivalent bottom angle is neglected (i.e., $\theta_{0}=0$), the parameters can be set as follows: $\omega_{0}=5000\;\mathrm{rad}/s$, $k_{1}=0.554$, $k_{2}=0.349$, and a=b=0.1. Then, a simulation can be performed for an input angular rate ranging from 0 to 2 rad/s. The result is shown in Figure 2. From the figure, it can be observed that when the input angular rate is 0 rad/s, the standing wave of the end face of the resonator’s lip is relatively static. However, when the input angular rate is increased to 2 rad/s, the waveform precesses in the opposite direction. This is the working principle of HRG, as the precession of the standing wave is proportional to the input angular rate and can be used to measure rotation.

## 3. Structural Types and Parameters of Hemispherical Resonator

There are three structural types based on the design of the hemispherical resonator. These three types are shown in Figure 3 and include the Ψ-type [16], Y-type, and mushroom-shaped type [17]. The Ψ-type and Y-type are suitable for the aHRG with a large radius, which uses a “three-piece set” or “two-piece set” spherical electrodes. On the other hand, the mushroom-shaped type is suitable for a small-radius planar electrode type HRG, where the lower end of the stem is fixed to the electrodes base. For this paper, the optimization object is a mushroom-shaped hemispherical resonator.

The key structural parameters of the mushroom-shaped hemispherical resonator include the mid-surface radius *R*, shell thickness *h*, stem diameter *D*, stem length *L*, fillet radii $R_{1}$ and $R_{2}$, top angle $\theta_{F}$, and equivalent bottom angle $\theta_{0}$. These parameters are illustrated in Figure 4a.

Equations (Equation 12) and (Equation 15) indicate that the 4-antinodes vibration mode frequency value is affected by material parameters such as Young’s modulus *E*, density ρ, and Poisson’s ratio μ, as well as structural parameters including the mid-surface radius *R*, shell thickness *h*, and equivalent bottom angle $\theta_{0}$. Additionally, the stem diameter *D* and fillet radii $R_{1}$ and $R_{2}$ are also important structural parameters that determine the equivalent bottom angle.

Sang-Jin Park et al. [14] conducted a study on the effect of the equivalent bottom angle on Bryan’s factor and proposed a more precise and comprehensive Bryan’s factor of hemispherical resonator expression. Their findings indicate that the equivalent bottom angle has a significant impact on Bryan’s factor. Specifically, when the stem diameter increases relative to the radius of the hemispherical shell, the influence of the stem on Bryan’s factor becomes more prominent.

Based on the given expression $f_{0}=\sqrt{k_{0}/m_{0}}/\left(2\pi \right)$, numerical calculations were carried out for different $\theta_{0}$ values under three different boundary conditions, and a simulation of a hemispherical shell without the stem was conducted. The results are presented in Figure 5. Figure 5a shows that under the boundary condition of w=0, the 4-antinodes vibration mode frequency value increases with the increase of $\theta_{0}$, eventually converging to a value. Under the boundary condition of v=0, the 4-antinodes vibration mode frequency value slightly decreases at first, then increases obviously, and reaches the minimum at $\theta_{0}=21^{{}^{\circ}}$. Under the boundary condition of u=0, the 4-antinodes vibration mode frequency value decreases with the increase of $\theta_{0}$. The simulation of the hemispherical shell without a stem shows that the 4-antinodes vibration mode frequency value is 5212 Hz (Figure 5b), and the theoretical value is 5841 Hz. The relative error is approximately 10.7%, which is within the error of 20% for the Kirchhoff–Love assumptions. It also proves the correctness of the theory. In theory, if the ratio of the thin shell’s thickness *h* to the mid-surface’s radius *R* is smaller, the relative error will also be smaller.

To investigate the specific impact of stem diameter *D* and fillet radii $R_{1}$, $R_{2}$ on the 4-antinodes vibration mode frequency value, this paper utilizes the classical parameters of the hemispherical resonator [4], with a mid-surface radius of R=15 mm and shell thickness of h=1 mm. The goal is to determine and optimize the stem diameter *D* as well as fillet radii $R_{1}$ and $R_{2}$.

## 4. Simulation and Determination of Response Indexes Based on Finite Element Method

### 4.1. Modal Simulation of Hemispherical Resonator

To perform modal analysis of a hemispherical resonator, we need to select appropriate material and structural parameters as shown in the Table 1 and use the finite element method. It is important to note that an ideal, defect-free hemispherical resonator should have evenly distributed circumferential properties, and the frequency of the same mode in different directions should be consistent. Therefore, ensuring a circumferential uniformity of grids and nodes is crucial when using the finite element method for modal analysis. To achieve this, we should manually mesh grids by using HyperMesh as shown in Figure 4b and avoid using automatic grid meshing. Additionally, for effective modal analysis, we need to fix the lower part of the stem by setting corresponding boundary conditions in the simulation software OptiSturct. Table 2 displays the first 10 modes’ frequency values, where the frequency of the same mode in different directions is consistent with four types in total: bending mode at the contact position between the stem and thin shell, 4-antinodes vibration mode (operational mode), stem bending mode, and 6-antinodes vibration mode, as shown in Figure 6. We refer to the 2nd and 3rd modes as the “pitching mode” and the 6th and 7th modes as the “bending mode”. The stem diameter *D* and fillet radii $R_{1}$, $R_{2}$ jointly affect the “pitching mode” and “bending mode”. Changing these three parameters may cause differences between the 4-antinodes vibration mode frequency value and the former and the later-order modes’ frequency values. In the HRG control system, it is important to ensure that the frequency of other modes that affect the operational mode are higher than the operational mode frequency as much as possible. Therefore, optimizing the parameters of the hemispherical resonator is necessary.

### 4.2. Determination of Response Indexes

The end face of the hemispherical resonator’s lip of planar electrode-type HRG forms capacitors with the planar electrodes. By applying the DC or AC voltage of a certain frequency and amplitude on the planar electrodes, the operational mode of the hemispherical resonator is excited, as shown in Figure 7a. To ensure the feasibility and stability of the control system, the frequency of the operational mode should be controlled between 5000 and 10,000 Hz and maintained at a certain difference from the frequency of other modes. Therefore, the 4-antinodes vibration mode frequency value is selected as the first response index $y_{1}$.

The hemispherical resonator is an axisymmetric oscillator with two 4-antinodes modes (modes 4, 5). Ideally, these two modes should have the same vibration frequency. However, due to defects in materials, processing, and manufacturing techniques, the produced resonator is not perfectly symmetrical in mass and stiffness distribution, which leads to frequency differences between the two 4-antinodes modes, which is known as frequency splitting. The lower frequency splits on the “heavy axis” (i.e., low-frequency stiffness axis), while the higher frequency splits on the “light axis” (i.e., high-frequency stiffness axis). The difference between these two frequencies is called the frequency splitting value, and the two stiffness axes are positioned 45° apart from each other. Frequency splitting is the main error source affecting the accuracy of the HRG, so it must be suppressed.

Using a simplified model of a ring resonator as an example, let the mass and resonant frequency of an ideal resonator be $M_{0}$ and ω, respectively. Then, *N* mass points are added to the ring resonator, where each mass point has a mass of $m_{i}$ and is positioned at $\varphi_{i}$. The positions of high-frequency and low-frequency axes are shown as $\phi_{1}$ and $\phi_{2}$, respectively, in Figure 7b.

According to the FOX’s theory [18,19], the position of the high-frequency stiffness axis is related to the uneven mass distribution and can be determined by:(22)$$\tan4\phi_{1}=\frac{\sum_{i=1}^{N}\left(m_{i}\sin4\varphi_{i}\right)}{\sum_{i=1}^{N}\left(m_{i}\cos4\varphi_{i}\right)}$$

The high- and low-frequency values are, respectively,
(23)$$\omega_{1}^{2}=\omega^{2}\left\{\frac{\left(1+\alpha_{2}^{2}\right)M_{0}}{\left(1+\alpha_{2}^{2}\right)M-\left(1-\alpha_{2}^{2}\right)\sum_{i=1}^{N}\left[m_{i}\cos\left(4\left(\varphi_{i}-\phi_{1}\right)\right)\right]}\right\}$$
(24)$$\omega_{2}^{2}=\omega^{2}\left\{\frac{\left(1+\alpha_{2}^{2}\right)M_{0}}{\left(1+\alpha_{2}^{2}\right)M+\left(1-\alpha_{2}^{2}\right)\sum_{i=1}^{N}\left[m_{i}\cos\left(4\left(\varphi_{i}-\phi_{1}\right)\right)\right]}\right\}$$
where $\alpha_{2}$ is the ratio between the radial and tangential amplitudes of the ring resonator in its 4-antinodes vibration mode, while *M* is the total mass of the ring resonator after adding *N* mass points. By using the two equations mentioned above, we can obtain:(25)$$\begin{array}{cc} \frac{\omega^{2}}{\omega_{2}^{2}}-\frac{\omega^{2}}{\omega_{1}^{2}} & =\frac{2\left(1-\alpha_{2}^{2}\right)\sum_{i=1}^{N}\left[m_{i}\cos\left(4\left(\varphi_{i}-\phi_{1}\right)\right)\right]}{\left(1+\alpha_{2}^{2}\right)M_{0}}\\ \; & =\frac{2\left(1-\alpha_{2}^{2}\right)}{\left(1+\alpha_{2}^{2}\right)M_{0}}\cdot \left[\sum_{i=1}^{N}m_{i}\cos4\varphi_{i}\cos4\phi_{1}+\sum_{i=1}^{N}m_{i}\sin4\varphi_{i}\sin4\phi_{1}\right]\\ \; & =\frac{\left(\omega_{1}+\omega_{2}\right)\left(\omega_{1}-\omega_{2}\right)\omega^{2}}{\omega_{1}^{2}\omega_{2}^{2}}\approx \frac{2\left(\omega_{1}-\omega_{2}\right)}{\omega } \end{array}$$
where the frequency-splitting value is $\omega_{1}-\omega_{2}=\lambda (\sum_{i=1}^{N}m_{i}\cos4\varphi_{i}\cos4\phi_{1}+$$\sum_{i=1}^{N}m_{i}\sin4\varphi_{i}\sin4\phi_{1})$. $\lambda =\left(1-\alpha_{2}^{2}\right)\omega /\left[\left(1+\alpha_{2}^{2}\right)M_{0}\right]$ is referred to as the mass sensitivity factor, which describes the influence level on frequency splitting when adding masses to the resonator. Its value is determined by the structure of the resonator. For this reason, the mass sensitivity factor λ is used as the second response index, and the lower its value, the better. For the purpose of simulating the effect of the mass sensitivity factor on frequency splitting, a hemispherical resonator with certain structural parameters was selected. The results are displayed in Figure 8. Based on Figure 8 and Equation (Equation 25), it can be observed that the frequency-splitting value is linearly proportional to the value of the circumferential uneven mass with the ratio being equivalent to the mass sensitivity factor. The mass sensitivity factor increases as the uneven mass becomes closer to the end face of the hemispherical resonator’s lip.

## 5. Experimental Design Based on Response Surface Method

### 5.1. Single-Factor Experiment

In order to investigate the influence of stem diameter *D* and fillet radii $R_{1}$ and $R_{2}$ on response indexes and determine the optimal range of three experimental factors’ levels, a single-factor experiment was conducted using the control variable method. The factor levels are shown in Table 3. Simulation results for the 4-antinodes vibration mode frequency value $y_{1}$ and mass sensitivity factor λ response indexes are presented in Figure 9, Figure 10 and Figure 11.

It is evident from Figure 9, Figure 10 and Figure 11 that when stem diameter *D* and fillet radius $R_{1}$ increase, both the 4-antinodes vibration mode frequency value and mass sensitivity factor increase significantly. With an increase in fillet radius $R_{2}$, the 4-antinodes vibration mode frequency value increases slightly, while the mass sensitivity factor exhibits a decreasing trend initially before increasing again with little clarity in the trend. The influence of experimental factors on the 4-antinodes vibration mode frequency value is consistent with the numerical calculations (w=0), indicating that the equivalent bottom angle affects the 4-antinodes vibration mode of the hemispherical resonator through radial constraints. Therefore, fillet radius $R_{2}$ is set to 0.8 mm with stem diameter *D* and fillet radius $R_{1}$ selected as experimental factors for the further optimization of structural parameters.

### 5.2. Central Composite Design

To research how stem diameter *D* and fillet radius $R_{1}$ jointly affect the 4-antinodes vibration mode frequency value and mass sensitivity factor, the central composite design method was utilized to optimize both parameters *D* and $R_{1}$. The factor levels are coded as shown in Table 4.

Table 5 presents the experiment schemes and simulation results, which underwent regression analysis. Table 6 and Table 7 display the analysis of variance (ANOVA) results. Upon analysis, it is found that the regression model for the 4-antinodes vibration mode frequency value $y_{1}$ is highly significant, with F-value = 97,959.40, *p*-value < 0.0001. Moreover, the lack of fit is not significant, with F_2_-value = 2.31, *p*${}_{2}$-value = 0.2150. Similarly, the regression model of mass sensitivity factor λ is also highly significant, with F-value = 51.09, *p*-value < 0.0001. Additionally, the lack of fit is not significant, with F_2_-value = 1.02, *p*${}_{2}$-value = 0.4924. Both regression equations exhibit a small proportion of abnormal errors, good fitting effects, and high reliability.

Regarding 4-antinodes vibration mode frequency value $y_{1}$, *D*, $R_{1}$, $DR_{1}$, $D^{2}$, $R_{1}^{2}$, and $DR_{1}^{2}$ are all terms that are highly significant (*p*-value < 0.01). For mass sensitivity factor λ, both *D* and $R_{1}$ are terms that are highly significant (*p*-value < 0.01), while $DR_{1}$ and $R_{1}^{2}$ are significant (0.01 < *p*-value < 0.05). By removing the insignificant terms from the regression equation of each experimental index, the resulting regression equation for the remaining terms with the 4-antinodes vibration mode frequency value $y_{1}$ and mass sensitivity factor λ is:(26)$$\begin{array}{cc} \; & y_{1}=6234.337+205.275R_{1}-329.029D-79.666DR_{1}-193.274R_{1}^{2}+34.124D^{2}+55.908DR_{1}^{2}\;\\ \; & \lambda =4.037-0.750R_{1}-0.011D+0.086DR_{1}+0.118R_{1}^{2} \end{array}$$

Figure 12 displays the response surfaces of stem diameter *D* and fillet radius $R_{1}$ to the 4-antinodes vibration mode frequency value $y_{1}$ and mass sensitivity factor λ. As shown in Figure 12, both the 4-antinodes vibration mode frequency value $y_{1}$ and mass sensitivity factor λ increase with an increase in stem diameter *D* and fillet radius $R_{1}$. When the stem diameter *D* is small, an increase in fillet radius $R_{1}$ results in a slow change in the 4-antinodes vibration mode frequency value $R_{1}$. Similarly, when fillet radius $R_{1}$ is small, an increase in stem diameter *D* leads to a slow change in the 4-antinodes vibration mode frequency value $y_{1}$. A similar trend can be observed for mass sensitivity factor λ.

## 6. Optimization of Equivalent Bottom Angle Parameters

Based on the machining and mechanical strength requirements, stem diameter *D* is selected to be more than 7 mm, and the fillet radius $R_{1}$ is between 1 and 2 mm. The 4-antinodes vibration mode frequency value ranges from 5000 to 10,000 Hz, and the mass sensitivity factor is set at its minimum value according to Equation (Equation 27). A numerical optimization was carried out, and the optimized parameters are shown in Table 8. For simulation, the following parameters are selected: *h* = 1 mm, *R* = 15 mm, D=7 mm, $R_{1}$ = 1 mm, and $R_{2}$ = 0.8 mm. The results show that the 4-antinodes vibration mode frequency value is 5441.761 Hz, and the mass sensitivity factor is 3.91 Hz/mg with relative errors of 0.13% and 0.51% with respect to the numerical optimized parameters shown in Figure 13, respectively, indicating that the regression model is correct. The frequency of the former and the latter mode of the 4-antinodes vibration mode are 5087.416 Hz and 9031.491 Hz, respectively, which will meet the excitation and working requirements perfectly.
(27)$$\begin{array}{c} \begin{array}{cc} \min & \lambda =f(D,R_{1})\\ s.t. & \left\{\begin{array}{c} D⩾7\\ 1⩽R_{1}⩽2\\ 5000⩽D⩽10,000 \end{array}\right. \end{array} \end{array}$$

## 7. Conclusions and Discussion

The kinetic equations of the hemispherical resonator were established using the Lagrangian method based on thin shell theory, and the working principle was simulated. The mushroom-shaped hemispherical resonator structural parameters were determined for a planar-electrode-type hemispherical resonator gyro. The effect of the equivalent bottom angle on the 4-antinodes vibration mode frequency value under different boundary conditions was investigated theoretically. The simulation results indicated that the equivalent bottom angle affects the 4-antinodes vibration mode of the hemispherical resonator through radial constraints.

A single-factor analysis was conducted on the equivalent bottom angle parameters that influence the 4-antinodes vibration mode frequency value and mass sensitivity factor, which include stem diameter *D*, fillet radii $R_{1}$ and $R_{2}$. The results indicate that an increase in stem diameter *D* and fillet radius $R_{1}$ leads to a significant increase in both the 4-antinodes vibration mode frequency value and mass sensitivity factor. However, the afillet radius $R_{2}$ has a minimal impact on the response indexes. A fillet radius of $R_{2}$ = 0.8 mm was chosen, and stem diameter *D* and fillet radius $R_{1}$ were used as experimental factors. A two-factor five-level central composite design and a response surface analysis were carried out. The results showed that the regression equation has a good fitting effect and high reliability. Both the 4-antinodes vibration mode frequency value $y_{1}$ and mass sensitivity factor λ increase with an increase in stem diameter *D* and fillet radius $R_{1}$.

The best optimized parameters of the hemispherical resonator, h=1 mm, R=15 mm, D=7 mm, $R_{1}=1$ mm, and $R_{2}=0.8$ mm, were selected and simulated. The results show that the 4-antinodes vibration mode frequency value is 5441.761 Hz, and the mass sensitivity factor is 3.91 Hz/mg with relative errors of 0.13% and 0.51% with respect to numerical optimized parameters.

At present, the mass sensitivity factor is based on the ring resonator as a theoretical model. However, in the future, the hemispherical resonator model will be utilized to analyze the impact of uneven mass on frequency splitting and to investigate how the structural parameters of the hemispherical resonator influence the frequency splitting specifically.

## Figures and Tables

**Figure 1 micromachines-14-01686-f001:**
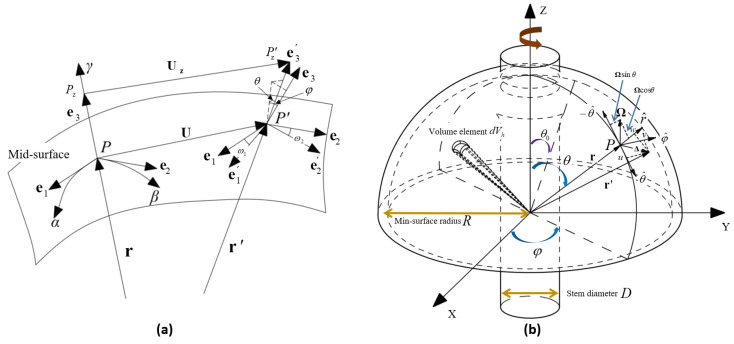
(**a**) Schematic diagram of shell deformation at a certain point; (**b**) Coordinate systems of the hemispherical resonator under rotation about Z axis.

**Figure 2 micromachines-14-01686-f002:**
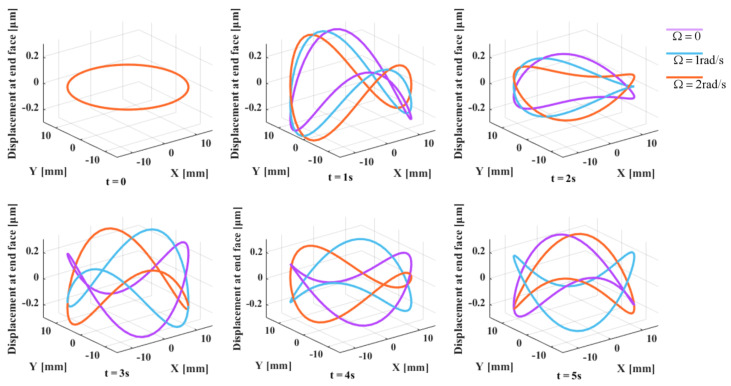
Phenomenon simulation of precession of hemispherical resonator.

**Figure 3 micromachines-14-01686-f003:**
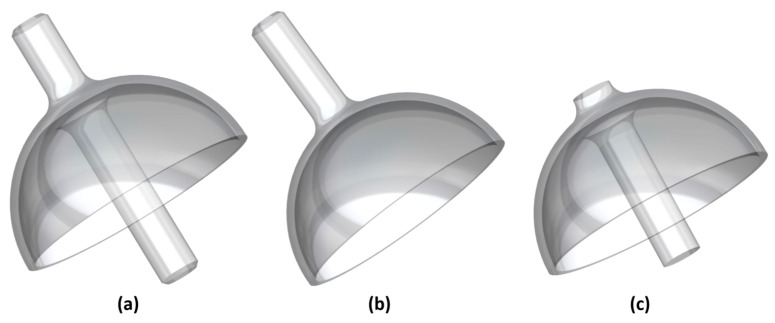
Structural types of hemispherical resonator, (**a**) Ψ-type, (**b**) Y-type, (**c**) mushroom-shaped type.

**Figure 4 micromachines-14-01686-f004:**
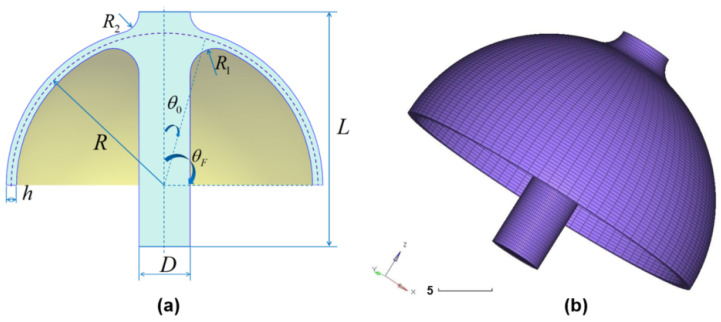
(**a**) Key structural parameters of the mushroom-shaped hemispherical; (**b**) Grids meshing of hemispherical resonator resonator.

**Figure 5 micromachines-14-01686-f005:**
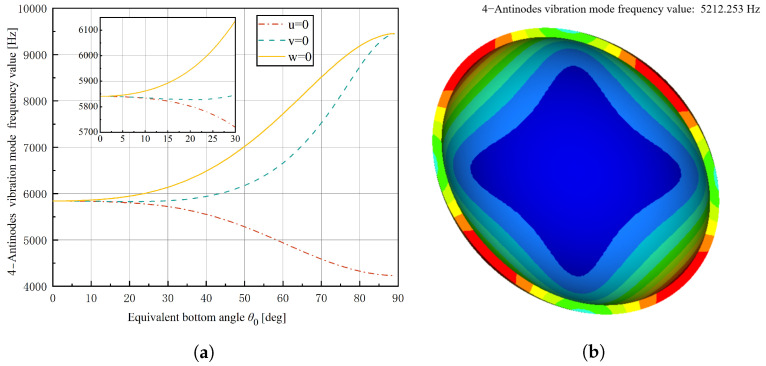
Theoretical values of 4-antinodes vibration mode frequency value for different boundary conditions and simulation of hemispherical shell, (**a**) Results of numerical calculations; (**b**) Simulation of 4-antinodes vibration mode of a hemispherical shell without steme.

**Figure 6 micromachines-14-01686-f006:**
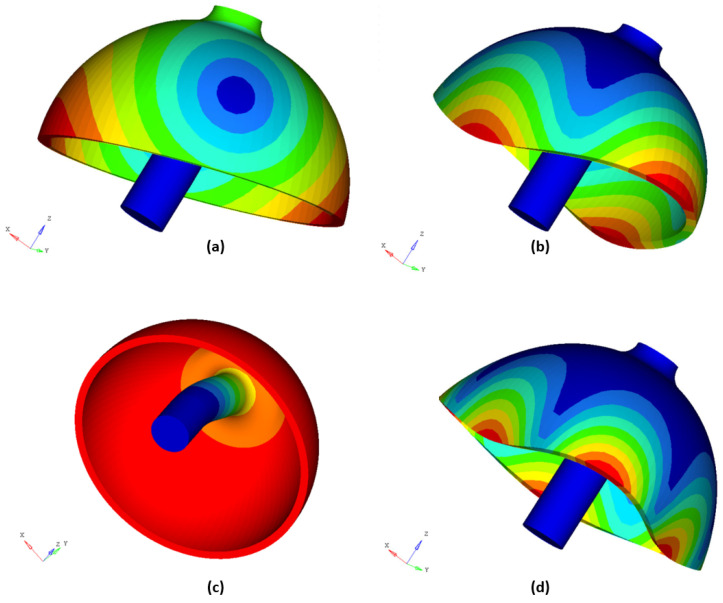
Vibration diagrams of 4 specific modes, (**a**) bending mode at the contact position between stem and thin shell, (**b**) 4-antinodes vibration mode, (**c**) stem bending mode, (**d**) 6-antinodes vibration mode.

**Figure 7 micromachines-14-01686-f007:**
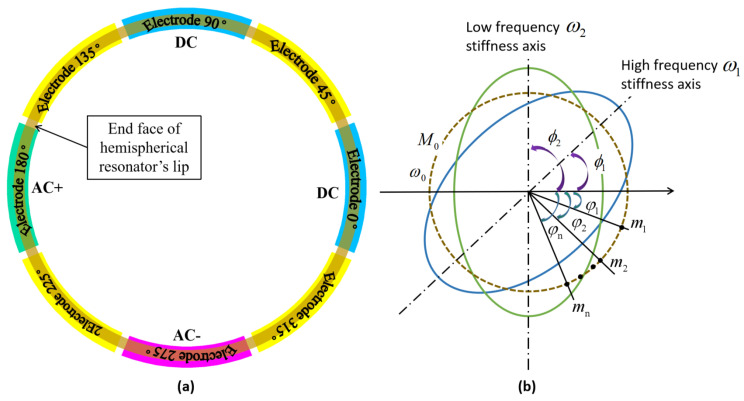
(**a**) Distribution of planar electrodes; (**b**) Schematic diagram of adding mass points.

**Figure 8 micromachines-14-01686-f008:**
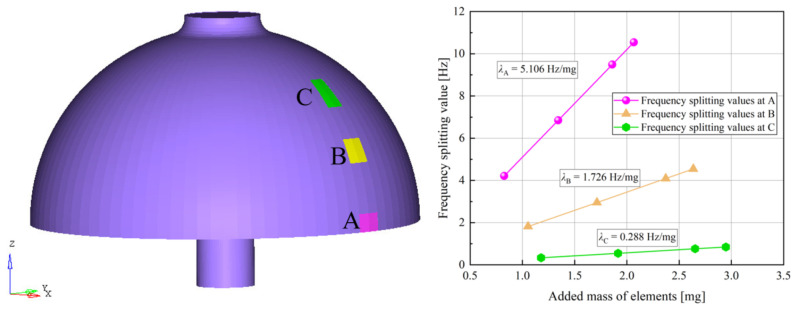
Simulation of frequency splitting and mass sensitivity factor.

**Figure 9 micromachines-14-01686-f009:**
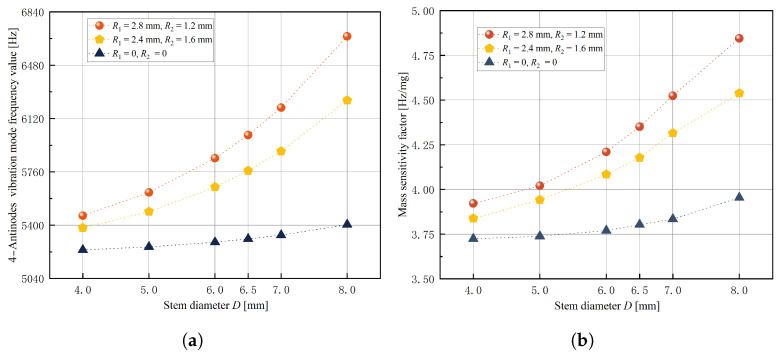
Simulation results of stem diameter *D*, (**a**) effect on 4-antinodes vibration mode frequency value $y_{1}$, (**b**) effect on mass sensitivity factor λ.

**Figure 10 micromachines-14-01686-f010:**
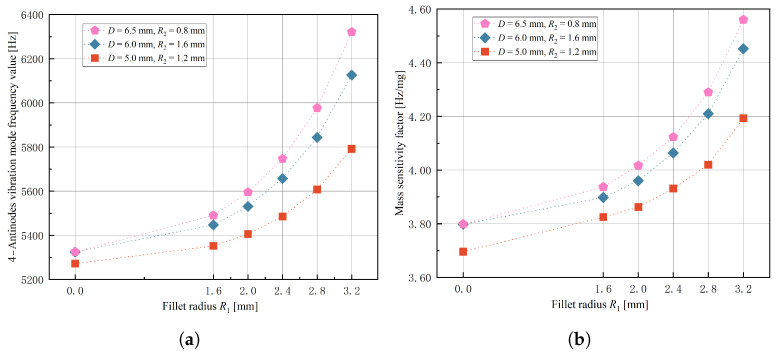
Simulation results of fillet radius $R_{1}$ , (**a**) effect on 4-antinodes vibration mode frequency value $y_{1}$, (**b**) effect on mass sensitivity factor λ.

**Figure 11 micromachines-14-01686-f011:**
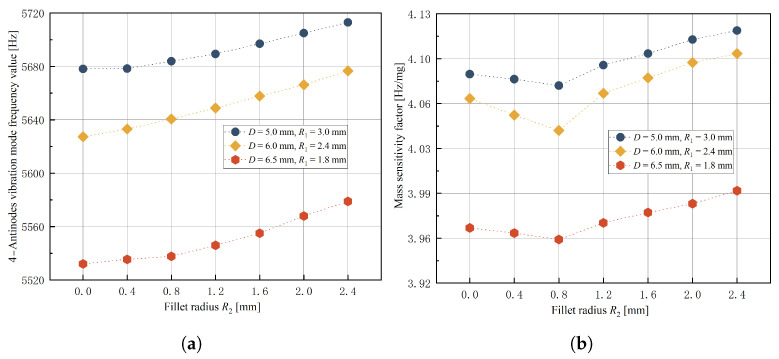
Simulation results of fillet radius $R_{2}$ , (**a**) effect on 4-antinodes vibration mode frequency value $y_{1}$, (**b**) effect on mass sensitivity factor λ.

**Figure 12 micromachines-14-01686-f012:**
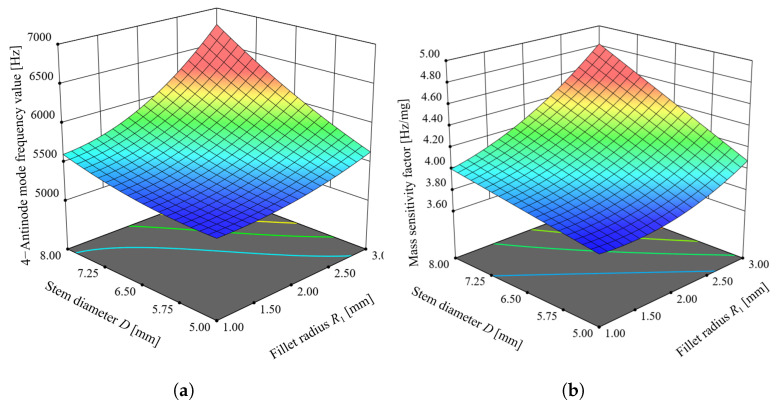
Response surface of experimental factors to response indexes, (**a**) effect of experimental factors on 4-antinodes vibration mode frequency value $y_{1}$, (**b**) effect of experimental factors on mass sensitivity factor λ.

**Figure 13 micromachines-14-01686-f013:**
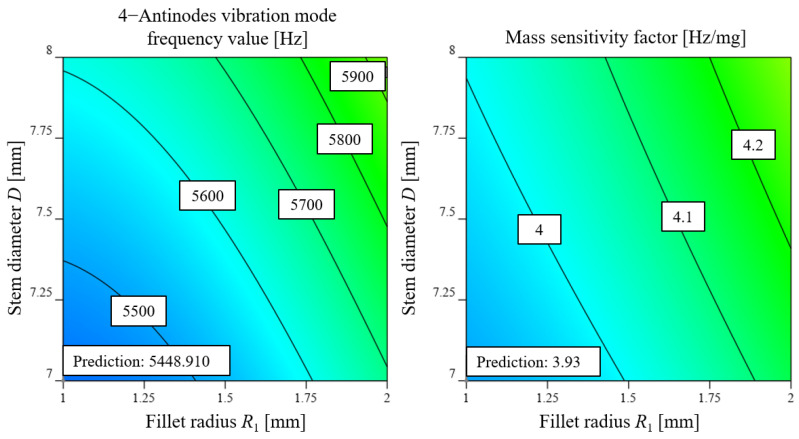
Prediction values of response indexes.

**Table 1 micromachines-14-01686-t001:** Material and structural parameters of hemispherical resonator.

Parameters	Values	Parameters	Values
Young’s modulus *E*	$7.67\times 10^{10}$ Pa	Mid-surface radius *R*	15 mm
Density ρ	$2200\;\mathrm{kg}/m^{3}$	Stem diameter *D*	5 mm
Poisson’s ratio μ	0.17	Fillet radius $R_{1}$	2.5 mm
Shell thickness *h*	1 mm	Fillet radius $R_{2}$	2 mm

**Table 2 micromachines-14-01686-t002:** Results of modal simulation.

Modes	Vibrations	Frequency Values (Hz)	Modes	Vibrations	Frequency Values (Hz)
1	Radial expansion mode of hemispherical shell	2850.397	6	Stem bending mode (along X direction)	6148.078
2	Bending mode at the contact position between stem and thin shell (along X direction)	3433.587	7	Stem bending mode (along Y direction)	6148.078
3	Bending mode at the contact position between stem and thin shell (along Y direction)	3433.587	8	6-Antinodes vibration mode (along X direction)	13,464.690
4	4-antinodes vibration mode (along X direction)	5523.907	9	6-Antinodes vibration mode (along Y direction)	13,464.690
5	4-antinodes vibration mode (along Y direction)	5523.907	10	Bouncing mode at the contact position between stem and thin shell	16,645.720

**Table 3 micromachines-14-01686-t003:** Levels of experimental factors.

Factors	Levels (mm)					
Stem diameter *D*	4.0	5.0	6.0	6.5	7.0	8.0
Fillet radius $R_{1}$	0	1.6	2.0	2.4	2.8	3.2
Fillet radius $R_{2}$	0	0.8	1.2	1.6	2.0	2.4

**Table 4 micromachines-14-01686-t004:** Factors level coding table.

Levels	Experimental Factors (mm)	
	Stem Diameter *D*	Fillet Radius *R*_1_
1.414	8.00	3.00
1	7.56	2.71
0	6.50	2.00
−1	5.44	1.29
−1.414	5.00	1.00

**Table 5 micromachines-14-01686-t005:** Experiment schemes and simulation results.

No.	Experimental Factors		Response Indexes	
	Stem Diameter D (mm)	Fillet Radius $R_{1}$ (mm)	4-Antinodes Vibration Mode Frequency Value $y_{1}$ (Hz)	Mass Sensitivity Factor λ (Hz/mg)
1	5.44	1.29	5344.95	3.807
2	5.44	2.71	5650.19	4.053
3	7.56	1.29	5567.77	4.074
4	7.56	2.71	6304.91	4.577
5	6.50	1.00	5394.84	3.863
6	6.50	3.00	6129.59	4.452
7	5.00	2.00	5400.54	3.882
8	8.00	2.00	5937.19	4.286
9	6.50	2.00	5593.32	4.007
10	6.50	2.00	5592.05	4.004
11	6.50	2.00	5593.12	3.984
12	6.50	2.00	5591.38	4.105
13	6.50	2.00	5594.10	4.081

**Table 6 micromachines-14-01686-t006:** ANOVA for reduced cubic model of 4-antinodes vibration mode frequency value $y_{1}$.

Source	Sum of Squares	df	Mean Square	F-Value	*p*-Value
	$9.803\times 10^{5}$	6	$1.634\times 10^{5}$	97,959.40	<0.0001
$R_{1}$	5.416 $\times \;10^{5}$	1	5.416 $\times \;10^{5}$	3.247$\times 10^{5}$	<0.0001
*D*	1.440 $\times \;10^{5}$	1	1.440 $\times \;10^{5}$	86,334.50	< 0.0001
$DR_{1}$	46,633.75	1	46,633.75	27,959.49	<0.0001
$R_{1}^{2}$	50,335.82	1	50,335.82	30,179.08	<0.0001
$D^{2}$	10,252.25	1	10,252.25	6146.79	<0.0001
$DR_{1}^{2}$	1758.20	1	1758.20	1054.14	<0.0001
Residual	10.01	6	1.67		
Lack of Fit	5.37	2	2.68	2.31	0.2150
Pure Error	4.64	4	1.16		
Cor Total	9.803 $\times \;10^{5}$	12			

Note: *p*-value < 0.01 (Highly significant), 0.01 < *p*-value < 0.05 (Significant), *p*-value > 0.05 (not significant).

**Table 7 micromachines-14-01686-t007:** ANOVA for reduced quadratic model of mass sensitivity factor λ.

Source	Sum of Squares	df	Mean Square	F-Value	*p*-Value
	0.5860	4	0.1465	51.09	<0.0001
$R_{1}$	0.3128	1	0.3128	109.08	<0.0001
*D*	0.2320	1	0.2320	80.90	<0.0001
$DR_{1}$	0.0165	1	0.0165	5.76	0.0432
$R_{1}^{2}$	0.0247	1	0.0247	8.61	0.0189
Residual	0.0229	8	0.0029		
Lack of Fit	0.0116	4	0.0029	1.02	0.4924
Pure Error	0.0114	4	0.0028		
Cor Total	0.6090	12			

Note: *p*-value < 0.01 (Highly significant), 0.01 < *p*-value < 0.05 (Significant), *p*-value > 0.05 (not significant).

**Table 8 micromachines-14-01686-t008:** Results of numerical optimization.

No.	Stem Diameter *D* (mm)	Fillet Radius $R_{1}$ (mm)	4-Antinodes Vibration Mode Frequency Value $y_{1}$ (Hz)	Mass Sensitivity Factor λ (Hz/mg)	Desirability	
1	7.000	1.000	5448.910	3.930	0.833	Selected
2	7.000	1.021	5449.916	3.932	0.830	
3	7.065	1.000	5457.163	3.935	0.827	
4	7.084	1.000	5459.650	3.936	0.825	

## Data Availability

Not applicable.

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
