# Peer review of "Simulation and Optimization of Hemispherical Resonator’s Equivalent Bottom Angle for Frequency-Splitting Suppression"

_micromachines, 2023, doi:10.3390/mi14091686_

Round 1
Reviewer 1 Report
Dear Author,
the paper is written very well. The Quality of the paper could be increased by adding some more references. First, there are some strong statements in the introduction section, which should be underlined by more than one reference. Second, beside the older references, which are very good, there should also be used some new references e.g.:
Song et al.: Research of Frequency Splitting Caused by Uneven Mass of Micro-Hemispherical Resonator Gyro; Micromachines 13(11):2015
Standing Wave Binding of Hemispherical Resonator Containing First–Third Harmonics of Mass Imperfection under Linear Vibration Excitation; Sensors19: 5454, 2020.
Author Response
Manuscript ID: micromachines-2526398
Type of manuscript: Article
Title: Simulation and Optimization of Hemispherical Resonator’s Equivalent Bottom Angle Based on Frequency Splitting Suppression
Comments and Suggestions for Authors
Dear Author,
the paper is written very well. The Quality of the paper could be increased by adding some more references. First, there are some strong statements in the introduction section, which should be underlined by more than one reference. Second, beside the older references, which are very good, there should also be used some new references e.g.:
Song et al.: Research of Frequency Splitting Caused by Uneven Mass of Micro-Hemispherical Resonator Gyro; Micromachines 13(11):2015
Standing Wave Binding of Hemispherical Resonator Containing First–Third Harmonics of Mass Imperfection under Linear Vibration Excitation; Sensors19: 5454, 2020.
Author Rebuttals to Initial Comments:
Reply: We thank the reviewer for appreciating the novelty of our approach and for the thoughtful comments about the true nature of our work.
We have carefully read the references you provided and cited them in our introduction section. In addition, in the introduction section, we have added the importance and necessity of parameter optimization, as follows:
In summary, when aiming to produce a hemispherical resonator, precise parameters for designing the resonator prove elusive, save for the constraints imposed by the manufacturing process. Consequently, in advancing the development of the hemispherical resonator gyro, the identification of suitable resonator parameters within specific limitations becomes pivotal.
Thank you again for your valuable opinion!

Reviewer 2 Report
The manuscript presented the simulation and optimization of the equivalent bottom angle of hemispherical resonators for suppression of frequency splitting. Based on thin shell theory, numerical models for simulating the kinetic behavior of hemispherical resonators have been established and is applied to study the effect of equivalent bottom angle parameters on the frequency of vibration modes. The paper has been well written and well organized. Thus, it is recommended for publication in Micromachines after minor revision.
It is suggested to add some discussion on the restrictions of the thin-shell theory based numerical models in the simulation of the kinetic behavior of hemispherical resonators.
The font sizes in Figure 3 and Figure 8 are too small. These two figures need to be enlarged to show text and numbers more clearly.
The English language of the manuscript need further improved. For example, the title of manuscript is confusing and needs rephrased, e.g. changing “Based on” to “for”.
Author Response
Manuscript ID: micromachines-2526398
Type of manuscript: Article
Title: Simulation and Optimization of Hemispherical Resonator’s Equivalent Bottom Angle Based on Frequency Splitting Suppression
Comments and Suggestions for Authors
The manuscript presented the simulation and optimization of the equivalent bottom angle of hemispherical resonators for suppression of frequency splitting. Based on thin shell theory, numerical models for simulating the kinetic behavior of hemispherical resonators have been established and is applied to study the effect of equivalent bottom angle parameters on the frequency of vibration modes. The paper has been well written and well organized. Thus, it is recommended for publication in Micromachines after minor revision.
It is suggested to add some discussion on the restrictions of the thin-shell theory based numerical models in the simulation of the kinetic behavior of hemispherical resonators.
The font sizes in Figure 3 and Figure 8 are too small. These two figures need to be enlarged to show text and numbers more clearly.
The English language of the manuscript need further improved. For example, the title of manuscript is confusing and needs rephrased, e.g. changing “Based on” to “for”.
Author Rebuttals to Initial Comments:
The manuscript presented the simulation and optimization of the equivalent bottom angle of hemispherical resonators for suppression of frequency splitting. Based on thin shell theory, numerical models for simulating the kinetic behavior of hemispherical resonators have been established and is applied to study the effect of equivalent bottom angle parameters on the frequency of vibration modes. The paper has been well written and well organized. Thus, it is recommended for publication in Micromachines after minor revision.
Reply:We thank the reviewer for appreciating the novelty of our approach and for the thoughtful comments about the true nature of our work.
It is suggested to add some discussion on the restrictions of the thin-shell theory based numerical models in the simulation of the kinetic behavior of hemispherical resonators.
Reply:We agree with this valuable comment. In fact, on lines 86 -95 of the manuscript, we have discussed the restrictions of the thin shell theory. On the other hand, after simulation, on line 162 of the manuscript, we compared the theoretical value with simulation value, indicating that it is within the range of error for the Kirchhoff-Love’s assumptions.
According your comment, we add some explanations about the restrictions of thin shell theory as follows: The relative error is approximately 10.7%, which is within the error of 20% for the Kirchhoff-Love’s assumptions. It also proves the correctness of the theory. In theory, if the ratio of the thin shell's thickness h to the mid-surface's radius R is smaller, the relative error will also be smaller.
The font sizes in Figure 3 and Figure 8 are too small. These two figures need to be enlarged to show text and numbers more clearly.
Reply:We have resized all the images so that the text on them can be clearly seen.
The English language of the manuscript need further improved. For example, the title of manuscript is confusing and needs rephrased, e.g. changing “Based on” to “for”.
Reply: We have refined the English language carefully. And we changed the title following the suggestion.

Reviewer 3 Report
This article presents the results of modeling the resonant frequencies of a fused quartz hemispherical resonator depending on its geometric parameters. For the selected resonator design, the authors calculated the frequencies of the lower 10 vibration modes. Calculations also showed that an increase in the stem diameter and fillet radii lead to an increase in the resonant frequency of the shell vibrations and the mass sensitivity factor. The authors determined the optimal stem diameter and fillet radii for a hemispherical resonator with the diameter of 15 mm and the wall thickness of 1 mm. The authors propose to use the created model in the future to analyze the effect of the inhomogeneous mass distribution on the frequency splitting.
Remarks:
1. It is not clear from the manuscript exactly how the calculations were carried out, with the help of what software. The authors presented calculated data for 10 lower modes of the resonator oscillations, but experimental data are not available. This makes it impossible to evaluate the accuracy of the finite element calculations.
2. In Fig. 14b, the dependence of the mass sensitivity factor on the fillet radius R2 has an extremum, it is not clear what it is connected with. Here, a calculation error is possible, since for small fillet radii, finite element calculation methods give a significant error. In any case, it was necessary to make sure that this extremum really exists, and to do this, carry out a similar calculation for at least one more point, for example, for R2=0.6 mm.
3. The authors encode the dimensions of the resonator in Table 4, then they indicate the codes in Table 5. This is inconvenient for reading, it should indicate the specified dimensions directly in Table 5.
4. The simulation results (Fig. 15) show that the resonant frequency and the mass sensitivity factor almost equally depend on the stem diameter and the radii of the surface rounding between the stem and the shell. This is a rather obvious result, taking into account the provisions of Fox's theory (Ref.[16,17]) and the relation given by the authors on line 219 of the manuscript, from which it follows that the mass sensitivity factor is linearly related to the resonant frequency. When optimizing the resonator parameters (D, R1) to minimize the mass sensitivity factor, the authors essentially chose D, R1 to achieve the minimum resonant frequency. This result does not contain novelty, since such finite element calculations of the resonant frequencies of a hemispherical resonator, as a rule, are always carried out during its design.
5. Although the authors indicate that they intend to use the created model in the future to analyze the effect of an inhomogeneous mass distribution on frequency splitting, this intention should be explained in more detail, using the example of one of the geometric parameters of the resonator. For example, from Table 8 it follows that an increase in the stem diameter from 7.065 mm to 7.084 mm and at R1=const leads to an increase in the resonant frequency of the resonator by approximately 2.5 Hz. Is it possible, on the basis of the created model, to determine the resulting frequency splitting if the cross section of the stem is not a circle, but an ellipse? Then, on the basis of such a model, it would be possible to determine the tolerances for the main dimensions of the resonator.
Conclusion: The manuscript needs significant revision.
Author Response
Manuscript ID: micromachines-2526398
Type of manuscript: Article
Title: Simulation and Optimization of Hemispherical Resonator’s Equivalent Bottom Angle Based on Frequency Splitting Suppression
Comments and Suggestions for Authors
This article presents the results of modeling the resonant frequencies of a fused quartz hemispherical resonator depending on its geometric parameters. For the selected resonator design, the authors calculated the frequencies of the lower 10 vibration modes. Calculations also showed that an increase in the stem diameter and fillet radii lead to an increase in the resonant frequency of the shell vibrations and the mass sensitivity factor. The authors determined the optimal stem diameter and fillet radii for a hemispherical resonator with the diameter of 15 mm and the wall thickness of 1 mm. The authors propose to use the created model in the future to analyze the effect of the inhomogeneous mass distribution on the frequency splitting.
Remarks:
- It is not clear from the manuscript exactly how the calculations were carried out, with the help of what software. The authors presented calculated data for 10 lower modes of the resonator oscillations, but experimental data are not available. This makes it impossible to evaluate the accuracy of the finite element calculations.
- In Fig. 14b, the dependence of the mass sensitivity factor on the fillet radius R2 has an extremum, it is not clear what it is connected with. Here, a calculation error is possible, since for small fillet radii, finite element calculation methods give a significant error. In any case, it was necessary to make sure that this extremum really exists, and to do this, carry out a similar calculation for at least one more point, for example, for R2=0.6 mm.
- The authors encode the dimensions of the resonator in Table 4, then they indicate the codes in Table 5. This is inconvenient for reading, it should indicate the specified dimensions directly in Table 5.
- The simulation results (Fig. 15) show that the resonant frequency and the mass sensitivity factor almost equally depend on the stem diameter and the radii of the surface rounding between the stem and the shell. This is a rather obvious result, taking into account the provisions of Fox's theory (Ref.[16,17]) and the relation given by the authors on line 219 of the manuscript, from which it follows that the mass sensitivity factor is linearly related to the resonant frequency. When optimizing the resonator parameters (D, R1) to minimize the mass sensitivity factor, the authors essentially chose D, R1 to achieve the minimum resonant frequency. This result does not contain novelty, since such finite element calculations of the resonant frequencies of a hemispherical resonator, as a rule, are always carried out during its design.
- Although the authors indicate that they intend to use the created model in the future to analyze the effect of an inhomogeneous mass distribution on frequency splitting, this intention should be explained in more detail, using the example of one of the geometric parameters of the resonator. For example, from Table 8 it follows that an increase in the stem diameter from 7.065 mm to 7.084 mm and at R1=const leads to an increase in the resonant frequency of the resonator by approximately 2.5 Hz. Is it possible, on the basis of the created model, to determine the resulting frequency splitting if the cross section of the stem is not a circle, but an ellipse? Then, on the basis of such a model, it would be possible to determine the tolerances for the main dimensions of the resonator.
Conclusion: The manuscript needs significant revision.
Author Rebuttals to Initial Comments:
- It is not clear from the manuscript exactly how the calculations were carried out, with the help of what software. The authors presented calculated data for 10 lower modes of the resonator oscillations, but experimental data are not available. This makes it impossible to evaluate the accuracy of the finite element calculations.
Reply: We agree with this valuable comment. We utilize HyperMesh for meshing and perform finite element analysis using OptiStruct. We have incorporated this into our manuscript. The frequency values are reflected in the modified Fig.6.
- In Fig. 14b, the dependence of the mass sensitivity factor on the fillet radius R2 has an extremum, it is not clear what it is connected with. Here, a calculation error is possible, since for small fillet radii, finite element calculation methods give a significant error. In any case, it was necessary to make sure that this extremum really exists, and to do this, carry out a similar calculation for at least one more point, for example, for R2=0.6 mm.
Reply: We have introduced an additional data point to substantiate the presence of this extremum, which has been duly adjusted within the manuscript. We may not currently infer the existence of this extreme value through theoretical formulas now, but there invariably exists a point of extremity for R2 as both D and R1 undergo changes. This observed trend aligns precisely with our requirements.
- The authors encode the dimensions of the resonator in Table 4, then they indicate the codes in Table 5. This is inconvenient for reading, it should indicate the specified dimensions directly in Table 5.
Reply: we changed the table following the suggestion.
- The simulation results (Fig. 15) show that the resonant frequency and the mass sensitivity factor almost equally depend on the stem diameter and the radii of the surface rounding between the stem and the shell. This is a rather obvious result, taking into account the provisions of Fox's theory (Ref.[16,17]) and the relation given by the authors on line 219 of the manuscript, from which it follows that the mass sensitivity factor is linearly related to the resonant frequency. When optimizing the resonator parameters (D, R1) to minimize the mass sensitivity factor, the authors essentially chose D, R1 to achieve the minimum resonant frequency. This result does not contain novelty, since such finite element calculations of the resonant frequencies of a hemispherical resonator, as a rule, are always carried out during its design.
Reply: In our pertinent work, the mass sensitivity factor is indeed linearly correlated with the 4-antinodes vibration mode frequency value, as follows:
. It should be noted that the 4-antinodes vibration mode frequency value is determined by parameters of resonator such as R, h, D, R1 and R2, as shown as the above equation. Thus, we chose the optimized parameters D and R1 to minimize the mass sensitivity factor in order to suppress the frequency splitting of hemispherical resonator. Besides, in our objective function, the goal is the minimum of mass sensitivity factor. As shown as section 4.2, the 4-antinodes vibration mode frequency is a range between 5000Hz and 10000Hz rather than a specific numerical values. Thus, it is not contradictory.
Any finite element method will generate errors, and we will find change rules while reducing errors, rather than determining the value of a specific response index.
- Although the authors indicate that they intend to use the created model in the future to analyze the effect of an inhomogeneous mass distribution on frequency splitting, this intention should be explained in more detail, using the example of one of the geometric parameters of the resonator. For example, from Table 8 it follows that an increase in the stem diameter from 7.065 mm to 7.084 mm and at R1=const leads to an increase in the resonant frequency of the resonator by approximately 2.5 Hz. Is it possible, on the basis of the created model, to determine the resulting frequency splitting if the cross section of the stem is not a circle, but an ellipse? Then, on the basis of such a model, it would be possible to determine the tolerances for the main dimensions of the resonator.
Reply: This suggestion to provide a more detailed explanation of our intention to use the model for analyzing the effect of inhomogeneous mass distribution on frequency splitting is well-taken. The main purpose of this article is to determine the values of D, R1 and R2 for frequency splitting suppression, as these two values are not directly reflected in other literature. The current machining level error has reached 1um, but this value cannot be achieved in finite element methods considering both errors and mesh grids size. It is unlikely to machine a circle section into an ellipse section. What we are doing is to find change rules rather than determining the value of a specific response index.
Besides, your suggestion regarding using the model to determine tolerances for the main dimensions of the resonator is insightful. We will certainly explore this avenue and incorporate it into our future work. This approach aligns well with our aim to provide practical applications for the model beyond its theoretical aspects.

Round 2
Reviewer 3 Report
I am satisfied with the answers of the authors and the corrections made in the manuscript.